# Experimental evidence that changing beliefs about mask efficacy and social norms increase mask wearing for COVID-19 risk reduction: Results from the United States and Italy

Scott E. Bokemper[1,2], Maria Cucciniello[3,4], Tiziano Rotesi[5], Paolo Pin[6,7], Amyn A. Malik[8,9], Kathryn Willebrand[8,10], Elliott E. Paintsil[8,11], Saad B. Omer[8,9,10,12], Gregory A. Huber[1,13]*, Alessia Melegaro[4,14]

1 Institution for Social and Policy Studies, Yale University, New Haven, Connecticut, United States of America, 2 Center for the Study of American Politics, Yale University, New Haven, Connecticut, United States of America, 3 Business School, University of Edinburgh, Edinburgh, United Kingdom, 4 Dondena Centre for Research on Social Dynamics and Public Policies, Bocconi University, Milan, Italy, 5 Department of Economics, University of Lausanne, Lausanne, Switzerland, 6 Department of Economics and Statistics, Università di Siena, Siena, Italy, 7 Bocconi Institute for Data Science and Analytics (BIDSA), Bocconi University, Milan, Italy, 8 Yale Institute for Global Health, New Haven, Connecticut, United States of America, 9 Yale School of Medicine, New Haven, Connecticut, United States of America, 10 Yale School of Public Health, New Haven, Connecticut, United States of America, 11 Institution of Human Nutrition, Columbia University, New York City, New York, United States of America, 12 Yale School of Nursing, Orange, Connecticut, United States of America, 13 Department of Political Science, Yale University, New Haven, Connecticut, United States of America, 14 Social and Political Science Department, Bocconi University, Milan, Italy

* gregory.huber@yale.edu

**Data Availability Statement:** Anonymized data files and replication code have been posted to

## Abstract

In the absence of widespread vaccination for COVID-19, governments and public health officials have advocated for the public to wear masks during the pandemic. The decision to wear a mask in public is likely affected by both beliefs about its efficacy and the prevalence of the behavior. Greater mask use in the community may encourage others to follow this norm, but it also creates an incentive for individuals to free ride on the protection afforded to them by others. We report the results of two vignette-based experiments conducted in the United States (*n* = 3,100) and Italy (*n* = 2,659) to examine the causal relationship between beliefs, social norms, and reported intentions to engage in mask promoting behavior. In both countries, survey respondents were quota sampled to be representative of the country's population on key demographics. We find that providing information about how masks protect others increases the likelihood that someone would wear a mask or encourage others to do so in the United States, but not in Italy. There is no effect of providing information about how masks protect the wearer in either country. Additionally, greater mask use increases intentions to wear a mask and encourage someone else to wear theirs properly in both the United States and Italy. Thus, community mask use may be self-reinforcing.

Harvard Dataverse https://doi.org/10.7910/DVN/NZYWS5.

**Funding:** SEB and GAH acknowledge support from the Institution for Social and Policy Studies, the Center for the Study of American Politics, and the Tobin Center for Economic Policy at Yale University. PP acknowledges support from the Italian Ministry of Education Progetti di Rilevante Interesse Nazionale Grant 2017ELHNNJ. AM acknowledges support from Italian Ministry of Education Progetti di Rilevante Interesse Nazionale Grant 20177BRJXS and the European Research Council Consolidator Grant 101003183. AAM, SBO, EP, and KW were supported by Yale Institute for Global Health. The funders had no role in study design, data collection and analysis, decision to publish, or preparation of the manuscript.

**Competing interests:** The authors have declared that no competing interests exist.

## Introduction

The COVID-19 pandemic has caused more than 184 million cases resulting in at least 3.9 million deaths globally as of July 5th, 2021 [1]. To slow the spread of the virus, governments and public health officials have encouraged, and in many instances mandated, the use of masks or other face coverings in public spaces. The effectiveness of this strategy is determined by the prevalence of mask wearing in the population [2]. Recent research finds that wearing a mask may protect both the wearer and those around them from contracting COVID-19 [3–8]. Additionally, people who believe that masks are effective report that they are more likely to wear them [9].

But, persuasive communication experiments have found mixed positive [10–13] and null effects for whether messaging increases people's intentions to wear a mask [14, 15]. Several experiments that have found positive effects of persuasive health communication have identified that appealing to protecting others may be a particularly effective strategy. For example, a message that emphasized how COVID-19 is a threat to "your community" increased people's intentions to wear a mask [10]. Similarly, other work has found that inducing empathy for an individual who is particularly vulnerable to COVID-19 increased mask wearing intentions relative to an information only condition and an untreated control condition [13]. Notably, the information only condition produced a small, but statistically insignificant, increase in individuals' intentions to wear a mask compared to the untreated control. This suggests that purely informational messages may not be effective at changing behavioral intentions, though other work has found that an infographic that provides information about how masks slow the spread of COVID-19 changes beliefs about mask wearing [16].

Increasing mask use is an important public health strategy as the more people who wear masks while in public reduces the spread of the disease. Whether any given individual chooses to wear a mask is likely a function of the degree to which they believe that wearing a mask protects them or those around them. But, given that mask wearing is easily observable, the behavior of others is also likely to influence an individual's decision to wear a mask. As more people make the decision to wear a mask, it creates an incentive for individuals to free ride on the protection afforded to them by others. This could cause high mask compliance to unravel, thus making everyone less safe. Alternatively, a higher prevalence of mask use creates a stronger social norm that could induce even higher levels of compliance within the population.

Recent survey evidence from Germany shows that people who wear a mask view others who also do so more favorably than those who do not, which supports the idea that wearing a mask in public is a social norm [17]. Further, people who report that their friends and family wear masks at a greater rate are more likely to do so themselves [18]. This raises the possibility that the decision to wear a mask may be influenced by the behavior of others. However, given the novelty of mask wearing in public places, particularly in the United States and Italy, it is not clear ex ante whether the effects of social norms will outweigh the incentives to free ride.

We conducted two vignette-based survey experiments in two of the countries most affected by COVID-19, one in the United States and one in Italy, to better understand the causal relationship between 1) beliefs about mask efficacy, and 2) others' masking behavior and mask-relevant behaviors and attitudes. First, we examined whether experimentally inducing changes in beliefs about mask efficacy affects people's willingness to use them, ask others to use them properly, and judgment of those who do not. We separately examine beliefs about whether masks protect the wearer from sick people or whether masks protect one from spreading the disease to others. Second, we examined how experimentally manipulated prevalence of mask use changes these outcomes.

One experiment was fielded in the United States, which does not have a national mask mandate and where public support for mask wearing is highly variable. The other experiment was fielded in Italy, which prior to our experiment being fielded instituted a national outdoor face mask mandate in an effort to contain the spread of SARS-CoV-2 infection. Notably, these experiments were fielded in the fall of 2020 when people had been exposed to public health messaging for many months making it a particularly difficult test of whether our experimental treatments could change people's beliefs and behavioral intentions.

## Background

The COVID-19 pandemic created an urgent need to get people across the world to change their daily behaviors to prevent the spread of the disease. In addition to social distancing, public health officials recommended that people wear masks if they had to go out and interact with others. The intuition behind wearing masks to prevent the spread of COVID-19 is relatively straightforward. By wearing a mask, a sick individual reduces the amount of virus that they expel into the air round them. Similarly, wearing a mask provides some protection for a healthy individual from contracting the virus from those around them. Past research has found that masks serve as an effective barrier for reducing the amount of virus that an individual exhales and inhales [19–22]. Accordingly, people receive the most protection if they wear a mask and those around them do as well [23].

Countries and localities have taken variable approaches to mask use during the pandemic. Some places chose to mandate masks that require all citizens to wear a mask when they are in a public place (or at least certain public places), while others have left the decisions up to individuals to wear masks as they see fit. The goal of mandates is to increase the rate of mask use, but it is not necessarily the case that mandates are required to achieve high rates of mask use in the population. The variation in rates of mask wearing has allowed for the assessment of mask wearing as a policy tool for reducing cases and mortality. In the United States, states that mandated mask use experienced a reduction in their COVID-19 cases compared to those that did not [24] and those who adopted mask mandates also slowed their hospitalization rate [25]. Evidence from Germany also found a reduction in severe cases in regions that adopted masks earlier in the pandemic [5]. Similarly, countries with higher levels of mask use, either due to policy or voluntary compliance, experienced lower mortality rates [26].

Given that greater rates of mask use are associated with decreases in cases and mortality, what determines if people wear masks? One potential approach is informing people about how masks work to protect the wearer and those around them. Betsch et al. [16] randomly assigned survey respondents to either view an infographic that showed how masks prevent the spread of COVID-19 or not. Those who viewed the infographic, particularly those who reported being opposed government intervention to compel mask wearing, reported greater beliefs about masks protecting both themselves and others, as well as increased the beliefs about the general effectiveness of wearing masking. Increasing knowledge about the effectiveness of mask wearing is important because people who report believing masks are effective are also more likely to report wearing masks [9]. However, these studies do not isolate which component of this intervention is important: Is it the effect of changing beliefs about a mask protecting oneself or changing beliefs that masks protect others?

While information appears to increase beliefs about the effectiveness of masks, other work has found that information alone may not be enough to move behavioral intentions. Compared to a baseline control condition, an informational intervention increased respondent's willingness to wear a mask in the near future when they interact with others, though this difference was not statistically significant [13]. A notable design feature of this work is that the

outcome measure asked respondents about their likelihood of wearing a mask generally rather asking whether they would wear one in a specific context that has more well-defined features, like going over to someone's house.

Alternatively, persuasive messaging efforts have found considerable effects of content that emphasizes the importance of protecting others. Messages that emphasized taking action to keep "your community" safe or the "protect others" increased mask wearing intentions relative to control conditions respectively [10, 27]. This work also observed that appeals to protecting oneself are generally ineffective. Importantly, this intervention included factual information about how masks protect others, but also made an argument that wearing a mask was important to protect others. Other work found that highlighting the experience of a particularly vulnerable individual who experienced a severe case of COVID-19 increased mask wearing intentions [13].

The effectiveness of prosocial appeals suggests that masks are not thought of as an individual decision, but rather a collective action problem in which each individual's actions affect others and there are incentives to free ride on the protective behaviors of others. Given that mask wearing can easily be observed, people can assess the overall level of cooperation around them. On one hand, when mask wearing is high, people are afforded more protection by the mask wearing of other and may decide to free ride on the contributions of others. On the other hand, high levels of mask use may create a social norm that increases the likelihood that an individual adopts mask wearing behavior themselves. That is, when most people are wearing masks it creates a descriptive norm that shows what other people do in the same situation.

And indeed, past research has found that people who reported regularly wearing masks express more negative sentiments toward an individual who is not wearing a mask [17]. If broad mask use creates a social norm, it should also increase the likelihood that people enforce the norm by intervening in situations where someone is noncompliant [28]. More generally, social norms have been shown to affect health behaviors across a variety of contexts [e.g. 29–34, see also 35]. Focusing on social distancing to reduce the spread of COVID-19, Wu and Huber [33] analyze survey data and find that individuals in the US are more likely to practice social distancing when they report individuals in their social network are also doing so after accounting for other demographics. Focusing on mask wearing, Barcelo and Sheen [36] analyze a large-scale survey conducted in Spain and show that individuals are more likely to wear a mask when rates of mask wearing are higher among those they live near, even after accounting for other demographic characteristics and COVID-19 risk factors. These observational analyses provide suggestive evidence that individuals' own behavior is affected by the behavior of those around them, although such observational analysis is subject to standard concerns about omitted variables bias such as whether there are unmeasured interventions or covariates that explain both an individual's behavior and the behavior of those in their social or geographic community. We note that we use the term social norms to refer to descriptive norms, which characterize what other people are doing, rather than injunctive norms, which describe what one ought to do. Descriptive norms may undergird expectations about injunctive norms, in that people may believe they are more likely to be judged negatively for failing to follow a norm when more individuals follow the norm.

## Data and methods

We fielded two vignette-based survey experiments in two of the countries most affected by COVID-19: the United States and Italy. Both experiments were fielded using samples provided by the survey vendor Lucid. Subjects completed the survey online. The survey was programmed and data collected using the survey software Qualtrics. The surveys received ethics

approval from the Social Science, Behavioral, and Educational Research Institutional Review Board at Yale University and the Research Ethics Committee at Bocconi University. Participants were compensated for their participation, the studies did not involve deception, and written consent was obtained from each respondent prior to study enrollment.

The United States experiment was fielded between October 1 and October 21, 2020 with a total sample of 3,100 respondents. In the United States, Lucid employs quota-based sampling to match population benchmarks on age, race, gender, education, and income. Generally, respondents recruited through Lucid more closely resemble those in the American National Election Study than those from other common sources of data like Amazon Mechanical Turk [see 37].

The Italy experiment was fielded between October 22, 2020 and November 8, 2020 with a total sample of 2,659 respondents. For the Italian sample, the quota-based sampling was done to match the Italian population on age, gender, geographical area and education (see S1 Table for demographics in the US and Italy).

We asked respondents to read three vignettes and report their own intended behavior or their evaluation of the behavior of a third party. We conducted pretests in both the United States and Italy to assess whether people viewed the vignettes as realistic situations that people could find themselves in. In both pretests, respondents, on average, reported that the vignettes were largely realistic.

The complete experiment proceeded as follows. After providing informed written consent, respondents read an unrelated vignette to assess their attention. Those who did not answer a comprehension question correctly could not proceed with the study and exited the sample. Respondents then completed a variety of items asked as pre-treatment covariates. Several of the items asked about their experience with COVID-19 and their social distancing and masking behavior during the pandemic. Most relevant for our purposes here, we asked respondents about the frequency with which they wear a mask and the rate that they see others wearing masks in their community.

After this, subjects then read their randomly assigned mask efficacy treatment and answered a pair of questions about mask efficacy, described in greater detail below. Subjects were not allowed to advance the page with the treatment text until 20 seconds had elapsed. The post-treatment questions allow us to assess the efficacy of the treatments in affecting beliefs about the degree to which masks are self-protecting or other-regarding. Next, respondents read three experimentally manipulated vignettes and provided their response to each vignette. This design and the measurement of outcomes were pre-registered unless otherwise noted.

Our core randomized treatments allow us to test whether beliefs about mask efficacy and the mask wearing behavior of others as described in a vignette affect one's own mask wearing and one's willingness to encourage mask wearing by others. At the subject level, we randomized respondents at equal rates to a *Placebo Control*, where they read a message unrelated to the topic of the experiment, a *Masks Protect You* treatment that explains how masks stop some percentage of SARS-CoV-2 virus particles in the air from being inhaled by healthy people, and a *Masks Protect Others* treatment that explains how masks stop some percentage of SARS-CoV-2 virus particles from being expelled into the air by sick people (see Table 1 for full text). The content of the placebo control condition described the costs and benefits of bird feeding in the United States experiment and the features of a mocha coffee maker (moka pot) in the Italian experiment (see S1 Appendix). This randomization allowed us to understand if manipulating different beliefs about how masks work has distinct effects on intended behaviors.

After reading their assigned mask efficacy treatment, respondents were asked about their level of agreement with two statements about how masks might work to slow the spread of COVID-19. The statements read: "A mask reduces the risk that <u>sick people</u> spread COVID-19

**Table 1. Mask belief treatments.**

| Treatment | Treatment Text |
|---|---|
| Masks Protect You | Scientists have shown that wearing a face covering over your mouth and nose substantially reduces your risk of COVID-19 infection by decreasing the amount of virus that you inhale into your nasal passages and lungs. |
| | When a sick individual breathes, sneezes, or coughs, virus particles are expelled into the air where they can be inhaled by others. The more of the virus that makes it into your body, the greater your chances of getting sick with COVID-19. |
| | Scientists tested masks to see if they would stop virus particles from getting into a person's body. They found that an N95 respirator stops at least 95% of virus particles, while a simple cotton cloth mask covering your nose and mouth stops approximately 55% of virus particles. |
| Masks Protect Others | Scientists have shown that wearing a face covering over your mouth and nose substantially reduces your risk of spreading COVID-19 to others by decreasing the amount of virus that makes it into the air where it can infect other people. |
| | When a sick individual breathes, sneezes, or coughs, virus particles are expelled into the air where they can be inhaled by others. The more of the virus that makes it into someone's body, the greater their chances of getting sick with COVID-19. |
| | Scientists tested masks to see if they would stop virus particles from getting into the air around them. They found that an N95 respirator stops at least 85% of virus particles, while a simple cotton cloth mask covering your nose and mouth stops approximately 80% of virus particles. |

Randomized at the subject level. See S2 Appendix for the Control Condition.

into the air around them" and "A mask reduces the risk that healthy people get COVID-19 from the air around them". The former measured the belief that masks protect others from contracting COVID-19 and the latter measured the belief that masks protect the wearer from getting COVID-19 from those around them. Respondents reported their level of agreement on a five-point scale with anchors of "Strongly disagree" and "Strongly agree".

At the vignette level, we randomized the behavior of the other people described in the vignette to be either that all/almost all other people were wearing masks or none/very few people were wearing masks. This randomization allowed us to test whether the behavior of others caused free riding in situations involving masks or if high compliance with the descriptive social norm of wearing a mask increases the likelihood that an individual also complies with the norm.

Specifically, each respondent completed one each of a scenario about their a) OWN masking behavior, b) behavior toward OTHERS who are not properly wearing a mask, and c) evaluation of a THIRD PARTY who was not wearing their mask properly and someone who took an action in response to that. These evaluations were embedded in vignette settings about a) withdrawing money from an ATM, b) taking a walk in a public PARK, and c) going to a house MEETING in their neighborhood (see S1 Appendix). The content of these scenarios was chosen because they were relatable in the both the United States and Italy. The PARK scenario is an outdoor activity, the MEETING scenario is described as an indoor activity, and the ATM scenario could either be indoors or outdoors. Each respondent read one scenario in each of the three setting in a random order with each scenario and setting drawn without replacement. These randomizations, the flow of the experiment, and sample size by condition are shown in S2 Appendix.

In the OWN behavior scenarios, respondents were asked what they would do in a situation where they had forgotten a mask. The four response options were variations of: 1) continue the activity as normal, 2) continue the activity, but keep your distance from others, 3) abandon the activity or 4) go retrieve their mask and then proceed with the activity.

In the OTHER behavior scenarios they were asked what they would do if someone else was not wearing their mask properly. The four response options were variations of: 1) continue the

activity as normal, 2) continue the activity, but keep your distance, 3) abandon the activity, and 4) ask the person who is not wearing their mask properly to adjust it so it is covering their mouth and nose. By specifying that an individual is improperly wearing their mask, the vignettes make clear that the person has a mask with them, but this of course decreases the potential social cost of asking someone to change their behavior.

Finally, in the THIRD PARTY scenario, respondents read a scenario as an unaffected third party and reported their evaluation of others. Specifically, respondents were asked about their evaluation of someone who is not properly wearing a mask and their evaluation of a person who took an action in response to that, as well as what they believe the person who is not wearing a mask would think about the action that the other person took. In this third party scenario, the behavior of the person toward the third party was randomly assigned from 1) proceeding with the activity as usual, 2) proceeding with the activity, but keeping their distance from the person who was not wearing a mask, 3) stopping the activity, or 4) asking the person to fix their mask. For both the person who was not wearing a mask and for the person who took action, respondents were asked how much they believed each of the following words described that individual: intelligent, trustworthy, selfish, competent, aggressive, likeable, and reckless. Response options were a five-point scale ranging from "Not at all" to "Very". In the third party scenario, we also randomized the names and genders of the individuals involved in the scenario to avoid potential confounding effects of name and gender.

## Analysis

We use ordinary least squares (OLS) regression to estimate treatment effects in the analyses below. All outcomes are coded to range from 0 to 1 so we discuss the results in terms of percentage point changes. We estimate models independently for the US and the Italian experiments. We present the analysis of pooled data by scenario type, e.g. OWN, and included controls for individual context, e.g. ATM. We discuss dichotomized measures of mask promoting behavior in text and display alternative codings of the main dependent variables for the OWN and OTHER scenarios in Fig 1. We also report results by context in text. This analysis was pre-registered, but for ease of presentation we show the pooled results. All models were estimated with Huber-White standard errors including covariates that can be found in S3 Appendix.

## Results

### Baseline mask wearing behavior and beliefs

We begin by examining self-reported behavior about mask use and the mask use that respondents observed in their community (Table 2). For individual mask use, the Italian sample is significantly more likely to report wearing a mask when out in public than the American sample. Italians also report observing greater levels of mask use in their community than the American sample. Thus, these experiments also allow us to ascertain whether the effect of our experimental treatments are robust across contexts that vary considerably in baseline mask use.

We can also assess baseline beliefs about the degree to which masks protect the wearer and those around them by examining beliefs in the Placebo Control condition (Table 2). Comparing across countries, we observe similar levels of the belief that masks protect you in the control condition, but we find suggestive evidence that Italians may be more likely to believe that masks protect others than Americans at baseline, though this difference was not significant at conventional levels. Notably, baseline beliefs about masks protecting others are higher than beliefs about masks protecting the wearer in both countries which suggests that it may be more difficult to change these beliefs with our mask efficacy treatments.

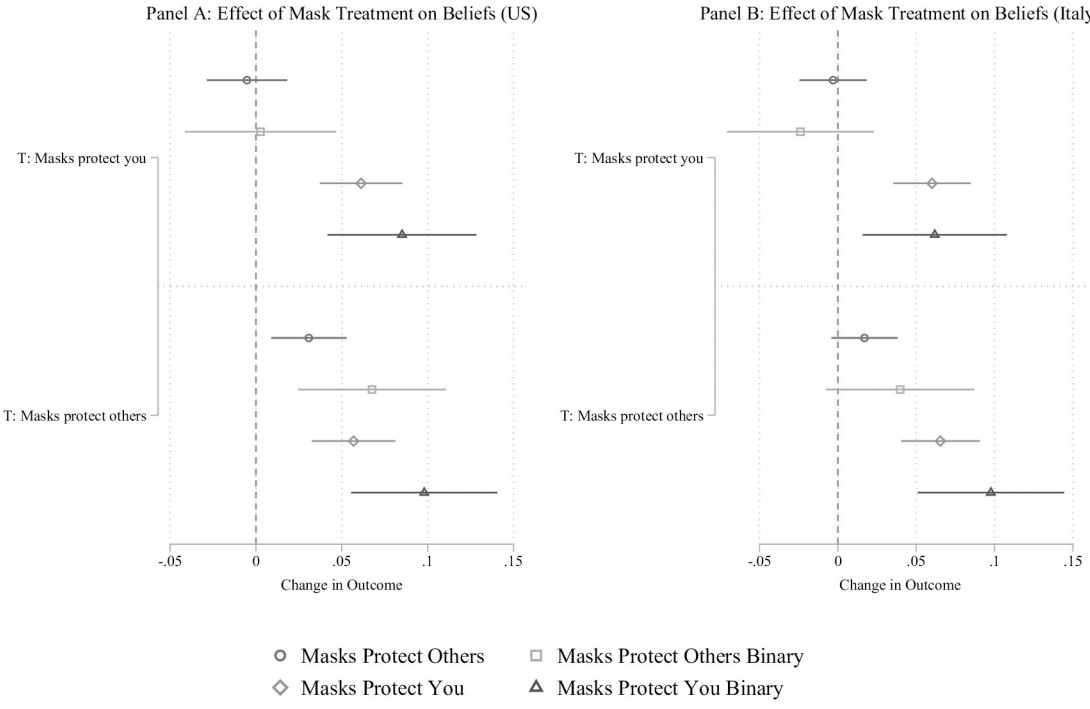

**Fig 1. Effect of information on beliefs about masks protecting you and protecting others compared to the control condition.** The figure displays OLS regression estimates with 95% confidence intervals. Models included covariates described above. For the binary outcomes, respondents who answered "strongly agree" were coded 1 and those who did not were coded 0.

## Effect of providing information about mask efficacy

Did providing information about how masks work affects beliefs about the efficacy of masks? For both countries, we validate that providing information changes beliefs about mask efficacy (see Fig 1 Panel A for the US and Panel B for Italy). Specifically, we observe that providing information about how masks protect the wearer increases beliefs that masks protect you without changing beliefs about whether masks protect others (effect on masks protect you in US is 6.1 percentage points, 95% C.I. = 3.7% to 8.5%, $p < .01$; effect in Italy is 6 percentage points, 95% C.I. = 3.5% to 8.4%, p < .01). Similarly, providing information that masks protect others

**Table 2. Baseline mask use and beliefs.**

|  | United States | Italy | T statistic (p) |
|---|---|---|---|
|  | Mean (S.D.) | Mean (S.D.) |  |
| Own Mask Use | 0.77 (0.31) | 0.89 (0.22) | 17.33 (< .001) |
| Mask Use Around You | 0.58 (0.32) | 0.74 (0.26) | 20.25 (< .001) |
| Masks Protect You (Control Group) | 0.74 (0.29) | 0.73 (0.28) | 1.10 (.31) |
| Masks Protect Others (Control Group) | 0.80 (0.26) | 0.82 (0.23) | 1.91 (.06) |

For own mask use and the observation of others mask use in daily life, Italians report that they are more likely to wear a mask and that they observe more mask wearing around them compared to Americans. Italian respondents and American respondents had similar baseline beliefs about the self-protecting effect of wearing a mask. We find suggestive evidence that Italian respondents were more likely to believe that masks also protect others in the control condition, although this difference was not significant at the conventional 5% level. Outcomes in the table are scaled to range from 0 to 1.

increases beliefs that masks protect others by 3.1 percentage points in the US (95% C.I. = 0.9% to 5.3%, $p < .01$). Although the effect is in the anticipated direction in Italy, it is not statistically different from 0 (the coefficient for Italy is 1.7%, 95% C.I. = -0.4% to 3.8%, p = .12). At the same time, information about how masks protect others also changes beliefs that masks protect the wearer. In practice, this means that we can perturb both sets of beliefs with information about masks protecting others, but can only change beliefs about masks protecting the wearer with that information.

Fig 2 Panels A and B plot the effect of the experimental interventions on reported behavior in the OWN behavior scenario pooling across settings for the US and Italy respectively. Fig 2 Panels C (US) and D (Italy) plot the same results in the OTHER behavior scenario. Panel A and Panel B provide clear evidence that providing information about how masks protect the wearer has no effect on own mask wearing behavior, whether measured using a scale (1 = Continue activity without a mask, 4 = go get a mask) or a binary outcome (1 = go get a mask, 0 = all other behavior). By contrast, providing information about how masks protects others increases reported risk reduction behavior in the US but not in Italy. For example, in the US sample, focusing on a dichotomous measure of mask wearing, the masks protect others intervention increases willingness to get a mask by 5.6 percentage points (95% C.I. = 1.4% to 9.7%, $p < .01$), an increase of 10.3% from the baseline level of 54.2% in the control group. Notably, we can likely attribute these effects to changes in beliefs about protecting others, because this is the only belief perturbed by the Masks Protect Others treatment that is not also affected by the Masks Protect You message. These effects are similar in magnitude among respondents who saw the OWN behavior scenario immediately after the mask efficacy treatment (see S4 Appendix).

Examining by scenario, the effect of the Masks Protect Others treatment was largest in the MEETING scenario, 6.8 percentage points (95% C.I. = 0.05% to 13.5%, $p < .05$), similar in magnitude to the pooled effect, though imprecise, for the ATM scenario, 5.3 percentage points (95% C.I. = -2.3% to 12.9%, $p = .17$), and smallest in the PARK scenario, 2.0 percentage points (95% C.I. = -5.3% to 9.3%, $p = .58$; See S5 Appendix for full results). In the Italian sample, the point estimates across all measures are negative, close to zero, and not statistically significant.

Results are similar in the Fig 2 Panel C and Panel D plot where the outcome is behavior toward the person who is not correctly wearing a mask: Providing information about how masks protect you has small and statistically insignificant effects, while providing information about how masks protect others increases both outcomes in the US but not it Italy. In the US, for the dichotomous willingness to ask another person to fix their mask outcome the effect is 4.4 percentage points (95% C.I. = 0.3% to 8.5%, $p < .05$), an increase of 13.6% from the baseline level of 32.5% in the control group. The estimates are less precise, but similar in magnitude, among respondents who saw the OTHER behavior scenario immediately after the mask efficacy treatment and at the scenario level (see S4 Appendix). The effect is again small and statistically insignificant in the Italian sample.

We display results from the third-party judgment scenario in Fig 3. Panels A and B plot evaluations of the person who is not correctly wearing their mask, while Panels C and D plot evaluations of the person who asks that person to fix their mask (Panels A and C for US and Panels B and D for Italy). For evaluations of a person who is not correctly wearing a mask, in neither sample is there an effect of either mask efficacy treatment on the judgment of the person incorrectly wearing their mask. Additionally, across countries, neither treatment has a statistically significant effect on judgments of the person who asks the person to fix their mask.

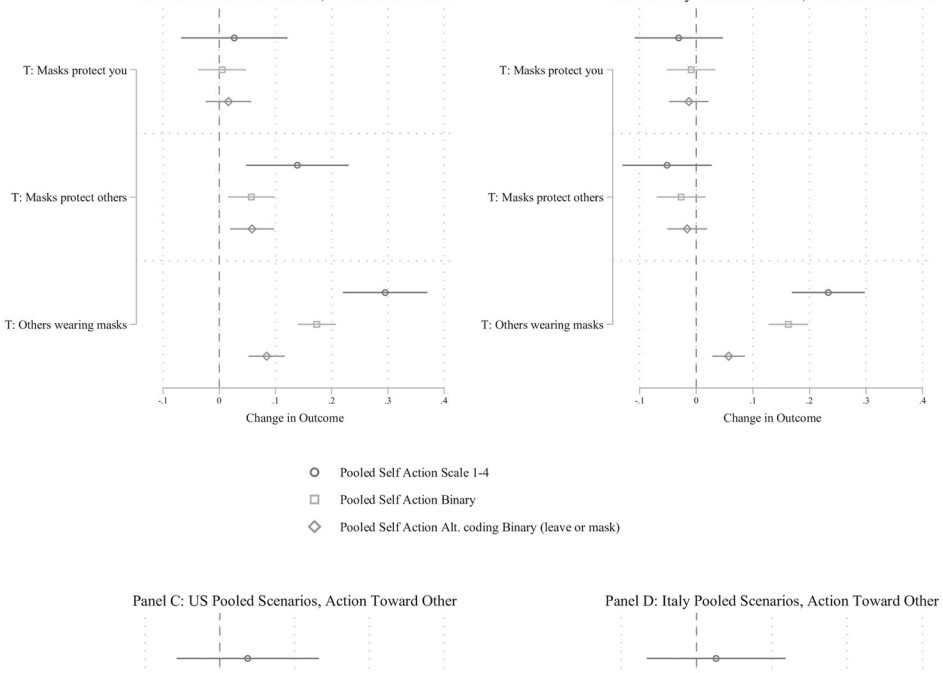

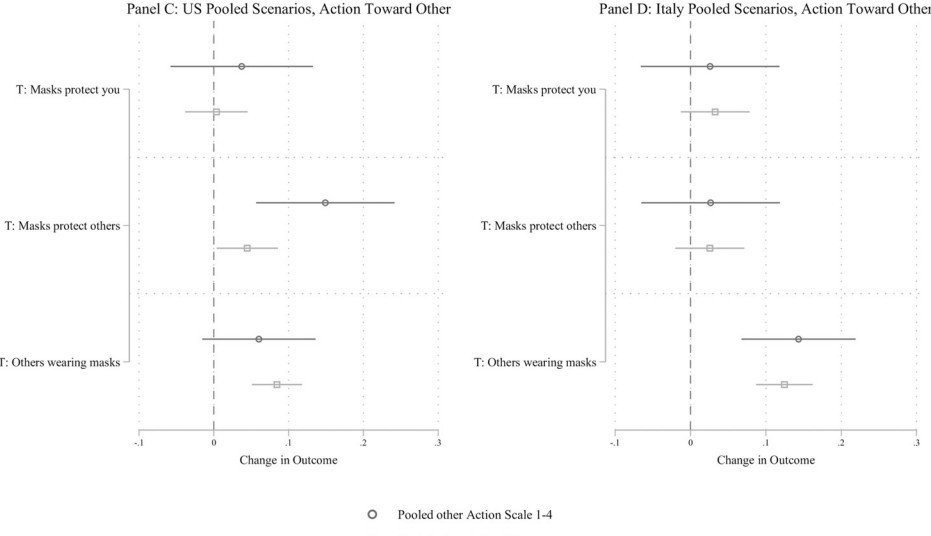

**Fig 2. Effect of beliefs and prevalence of mask wearing on behavioral intentions.** Panels A and B show the effect of the mask efficacy treatments and social norms treatment for the OWN mask behavior outcomes for the US and Italy respectively. Panels C and D show the same treatment effects for the OTHERS outcomes. The figure displays OLS regression estimates with 95% confidence intervals. Models include covariates described in S3 Appendix. For the United States, we observe a positive and statistically significant effect of the Masks Protect Others treatment on the OWN and OTHERS outcomes. For the others wearing masks treatment, participants in both the United States and Italy were more likely to report that they would retrieve their mask or ask someone else to fix their mask when most or all of the other people described in the vignette were wearing their mask.

## Effect of providing information about others' behavior

We next turn to examining the effect of the behavior of other people described in the vignettes, specifically whether others were wearing masks properly. As a reminder, it is unclear whether more prevalent mask wearing will increase or decrease others' willingness to wear masks or push others to do so. If others are wearing masks, the risks of contracting COVID-19 are already reduced and people may not wear a mask because they receive protection from others.

Additionally, the more people who are wearing masks, the more people who could intervene in the situation. Alternatively, if descriptive social norm effects are larger, others wearing masks could cause greater mask wearing and enhance individuals' willingness to act toward others.

Overall, the evidence strongly leads toward the latter norms interpretation. Returning to Panels A and B of Fig 2, the effect of other people wearing masks rather than not is to increase the likelihood someone returns to get their own mask by 17.3 percentage points (95% C.I. = 13.9% to 20.8%, $p < .01$) and 16.3 percentage points (95% C.I. = 12.8% to 19.8%, $p < .01$) for, respectively, the US and Italy, increases of 32% and 27% compared to baseline. Per Panels C and D, it also increases the willingness to ask others to fix their mask by 8.4 percentage points (95% C.I. = 5.1% to 11.8%, $p < .01$) in the US and 12.5 percentage points (95% C.I. = 8.7% to 16.2%, $p < .01$) in Italy, increases of 26% and 29.4% compared to baseline levels. Analyzing the data by specific scenario, the effect of other people wearing masks was positive and significant for both one's willingness to get their own mask and to ask others to fix their mask for all three scenarios in the US and in the ATM and MEETING scenarios in Italy.

This pattern is also apparent in the third-party evaluation scenario. Per Panel A and Panel B of Fig 3, a person who is not wearing their mask correctly is judged more negatively when more people are wearing their masks, although this effect is not statistically significant in the US (effect of .015 units in US, 95% C.I. = -.001 to .031, $p = .06$; effect of .02 units in Italy, 95% C.I. = .003 to .037, $p = .02$). Similarly, people judge a third party who asks someone else to fix their mask significantly less negatively in the US (B = -.035, 95% C.I. = -.005 to -.067, $p < .05$)

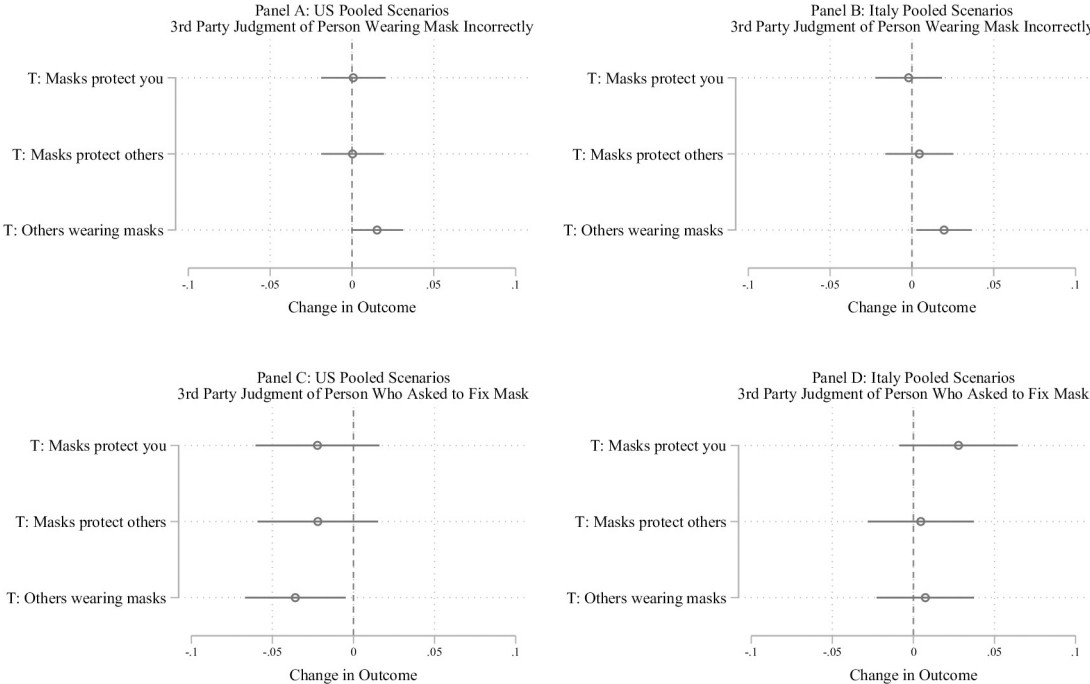

**Fig 3. Effect of beliefs and prevalence of mask wearing on third party judgment.** Panels A and B show the effect of the mask efficacy treatments and social norms treatment for judging someone who was not wearing their mask appropriate in the THIRD PARTY scenario. Panels C and D show the effect of these treatments for judging someone who asked a person to fix their mask in the THIRD PARTY scenario. The figure displays OLS regression estimates with 95% confidence intervals. Models include covariates described in S3 Appendix. Italian participants were more likely to judge negatively someone who was not wearing their mask properly when everyone else was wearing their mask properly, while participants in the United States viewed someone who asked another person to fix their mask more positively when everyone else was wearing their mask properly. The mask efficacy treatments did not have statistically significant effects on these outcomes.

when others are wearing their masks, with a null effect in the Italian sample. This link between judgement of others and descriptive norms highlights the way in which descriptive norms may undergird injunctive norms.

Overall, in describing their own masking behavior, their behavior toward others, and their evaluation of third parties, the normative effect of others' masking behavior on outcomes is clear across all of these scenarios in the US, but less clear in the Italian cases. Thus, despite concerns that more ubiquitous mask wearing might undercut further encouragement of masking, the effects are the opposite: Compliance by others appears to enhance both individuals' own behavior and support for behavior towards others. Importantly, there are no cases in which more frequent mask wearing depress others' willingness to wear masks or ask others to properly wear their masks.

## Discussion

Persuading individuals to engage in COVID-19 risk reduction is an important problem of public policy and public health. We demonstrate that in trying to encourage greater mask wearing, interventions that increase the belief that masks protect others appear promising, particularly in the United States. This suggests that factual information about how facemask use protect others are more effective than messages that emphasize how masks protect the wearer for increasing facemask use and encouraging others to do so. These effects are stronger in the US than in Italy, which may indicate that the small differences in beliefs at baseline were consequential or differences in how each sample is persuaded by these messages. Similarly, messages communicating social benefits and providing prosocial nudges have been shown to increase vaccination intentions in other studies[38, 39]. These social nudges have also been found to be effective in other contexts [40].

Additionally, we show that increased mask wearing may create self-reinforcing cycles that further promote the behavior. Increased mask wearing does not lead to free riding behavior or a willingness to defer to others in enforcing mask wearing. Instead, more frequent mask wearing creates a stronger descriptive social norm inducing better compliance with the mask wearing behavior in both the US and Italy. These findings are robust to various scenarios and vignettes tested in our experiment. Our results are consistent with those by Betsch et al. [38] and McKillop et al. [29] showing that emphasizing the high levels of neighborhood vaccination and the social benefits of vaccination did not induce free riding behavior. Other work has shown that peer influence can inform vaccination decisions, with these effects increasing as overall vaccination rates increase [32]. Our study suggests similar dynamics in an environment where peer influence is perhaps even more important, private mask-wearing behavior.

Our results were consistent in two independently conducted studies in two different settings: the US and Italy. This is important as the US did not have a facemask mandate while Italy did, increasing the generalizability of our results (although beliefs were harder to change in the Italian case). But both the US and Italy are high income western countries. Hence our findings need to be evaluated in other societies for applicability. However, previous work has shown that different cultures are receptive to prosocial messaging leading to increased intentions to perform the desired action, especially in individualistic societies [39].

We note four limitations of this study. First, our measures are not behavior, but are instead measures of behavioral intentions. The evidence presented here should be used for the design of field trials to validate these estimates in the non-survey setting and with broader populations. Second, our third-party enforcement scenarios involve situations where someone has a mask but is not wearing it correctly, a less difficult case than when someone confronts a third party who does not have a mask at all. Third, mask guidance has changed in both the United

States and Italy since the fielding of these studies, which may also change how people respond to others wearing masks. Last, we have not investigated how these effects are likely to change as vaccination against COVID-19 becomes more prevalent. It may be that the possibility someone is vaccinated will decrease the willingness to try to convince others to wear a mask because of ambiguity about whether another person is a risk factor. These questions are ripe for further investigation in follow-up work.

Our findings suggest first that public health communication campaigns that emphasize messages about how facemask use protect others are more likely to be effective in promoting mask wearing than campaigns that highlight how facemasks protect the wearer. The greater efficacy of this other-regarding message is clear in the United States, but this may not apply in other contexts as demonstrated by the results in Italy where neither mask efficacy treatment was effective. Additionally, we find that increasing mask use increases the willingness of others to wear masks and to encourage others to do so also, reducing concerns about how more prevalent mask wearing will lead to free-riding. The magnitudes of the effects of peer behavior are large and robust in both the American and the Italian context.

## Supporting information

**S1 Appendix. Mask efficacy treatments and vignettes.**
(DOCX)

**S2 Appendix. Progression of experiment and randomizations.**
(DOCX)

**S3 Appendix. Covariates and text of pre-treatment mask wearing items.**
(DOCX)

**S4 Appendix. First scenario robustness analysis.**
(DOCX)

**S5 Appendix. Scenario level analysis.**
(DOCX)

**S1 Table. Demographic information.**
(DOCX)

## Author Contributions

**Conceptualization:** Scott E. Bokemper, Maria Cucciniello, Tiziano Rotesi, Paolo Pin, Amyn A. Malik, Kathryn Willebrand, Elliott E. Paintsil, Saad B. Omer, Gregory A. Huber, Alessia Melegaro.

**Funding acquisition:** Paolo Pin, Saad B. Omer, Gregory A. Huber, Alessia Melegaro.

**Investigation:** Scott E. Bokemper, Maria Cucciniello, Tiziano Rotesi, Paolo Pin, Gregory A. Huber, Alessia Melegaro.

**Writing – original draft:** Scott E. Bokemper, Maria Cucciniello, Tiziano Rotesi, Paolo Pin, Gregory A. Huber.

**Writing – review & editing:** Scott E. Bokemper, Maria Cucciniello, Tiziano Rotesi, Paolo Pin, Amyn A. Malik, Kathryn Willebrand, Elliott E. Paintsil, Saad B. Omer, Gregory A. Huber, Alessia Melegaro.

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
