## [Decision Letter · Decision Letter 0]

16 Sep 2021

PONE-D-21-22869Experimental evidence that changing beliefs about mask efficacy and social norms increase mask wearing for COVID-19 risk reduction: Results from the United States and ItalyPLOS ONE

Dear Dr. Huber,

Thank you for submitting your manuscript to PLOS ONE. After careful consideration, we feel that it has merit but does not fully meet PLOS ONE’s publication criteria as it currently stands. Therefore, we invite you to submit a revised version of the manuscript that addresses the points raised during the review process.

We look forward to receiving your revised manuscript.

Kind regards,

Valerio Capraro

Academic Editor

PLOS ONE

Additional Editor Comments :

I have now collected two reviews from two experts in the field. Both reviews are very positive and suggest only a set of minor comments before publication. Therefore, I would like to invite you to revise the paper for Plos One. I am looking forward for the final version.

2. Please provide additional details regarding participant consent. In the Methods section, please ensure that you have specified (1) whether consent was informed and (2) what type you obtained (for instance, written or verbal). If your study included minors, state whether you obtained consent from parents or guardians. If the need for consent was waived by the ethics committee, please include this information.

“SEB and GAH acknowledge support from the Institution for Social and Policy Studies, the Center for the Study of American Politics, and the Tobin Center for Economic Policy at Yale University. PP and AM acknowledge support from the Italian Ministry of Education Progetti di Rilevante Interesse Nazionale (PRIN), respectively grant number 2017ELHNNJ and 20177BRJXS.

AAM, SBO, EP, and KW were supported by Yale Institute for Global Health.”

“SEB and GAH acknowledge support from the Institution for Social and Policy Studies, the Center for the Study of American Politics, and the Tobin Center for Economic Policy at Yale University. PP and AM acknowledge support from the Italian Ministry of Education Progetti di Rilevante Interesse Nazionale (PRIN), respectively grant number 2017ELHNNJ and 20177BRJXS.

AAM, SBO, EP, and KW were supported by Yale Institute for Global Health.”

“SEB and GAH acknowledge support from the Institution for Social and Policy Studies, the Center for the Study of American Politics, and the Tobin Center for Economic Policy at Yale University. PP and AM acknowledge support from the Italian Ministry of Education Progetti di Rilevante Interesse Nazionale (PRIN), respectively grant number 2017ELHNNJ and 20177BRJXS.

AAM, SBO, EP, and KW were supported by Yale Institute for Global Health.”

Reviewers' comments:

Reviewer's Responses to Questions

**Comments to the Author**

1. Is the manuscript technically sound, and do the data support the conclusions?

Reviewer #1: Yes

Reviewer #2: Yes

2. Has the statistical analysis been performed appropriately and rigorously? 

Reviewer #1: Yes

Reviewer #2: Yes

3. Have the authors made all data underlying the findings in their manuscript fully available?

Reviewer #1: Yes

Reviewer #2: Yes

4. Is the manuscript presented in an intelligible fashion and written in standard English?

Reviewer #1: Yes

Reviewer #2: Yes

5. Review Comments to the Author

Reviewer #1: This is an excellent paper that I enjoyed reading. It tackles an important question with both theoretical and practical implications. I have a few suggestions to strengthen the paper that need revising/clarifying before acceptance.

1. The authors focus the literature review on discussing prior experimental inducements of mask-wearing behavior. However, they do not incorporate some observational evidence that directly speaks to their paper. In a closely related paper published recently in PLOS ONE, Barcelo and Sheen (BS) provide observational evidence indicating “a positive correlation between a social norm of mask-wearing and mask uptake and demonstrate that uptake of facemasks is especially high among the elderly living in localities where mask-wearing behavior is popular” (1). In my opinion, the findings and interpretation of the results from this paper strongly resonate with those of BS.

Using evidence from a quota-based sample of 4,000 respondents living in Spain, BS find that mask-wearing behavior is correlated with the actual number of individuals wearing a mask in their location of residence. Quoting directly from the paper: “Table 4 reports ordinal logistic regressions that link the social acceptability of the mask-wearing behavior in a respondent’s social context and individual mask-wearing behavior after adjusting or not for demographic characteristics. The models reveal that respondents are more likely to wear a mask if they live in a region or a province where mask-wearing behavior is common. By contrast, the actual number of infected cases in a province or a region is not systematically associated with using a mask. The significant effect of social acceptability upholds after controlling for the number of infected cases in the region or province and adding fixed effect by survey day” (10-11).

Due to the resemblance of the main findings and interpretation of the results between these two papers, the authors must engage and discuss their evidence with regards to this paper in either the Introduction or the Discussion section.

With this comment, I do not want to suggest that this paper is not sufficiently novel. On the contrary, I believe that the authors offer an extension that critically contributes to our pre-existing knowledge by offering experimental evidence on an important research question. Overall, this paper offers much targeted, improved, and strengthened evidence on the role of social norms on mask-wearing behavior.

2. The authors may want to explicitly clarify that their paper looks at the effect of descriptive, rather than injunctive, social norms. In the authors’ social norms manipulation, the authors randomly assign respondents one of the two following conditions: “Social Norms Manipulation: Not many of the other people at the park are properly wearing a face mask OR Almost all of the other people at the park are properly wearing a face mask.” (Appendix S1).

Their experimental treatment deals with people’s perception of the prevalent behavior in their surroundings, not so much about what others approve or disapprove of. While the term “descriptive social norm” is used once in the paper, “social norm” as such is used about 20 times. I think that it is worth clarifying this concept early on to avoid confusion among readers and clarify the scope conditions of the paper.

Reviewer #2: The article addresses the important topic of whether changing beliefs about mask efficacy and social norms increase mask wearing. A particular strength of the text is the rigorously conducted studies with representative samples in two cultural contexts. Unlike many existing studies, the results obtained appear to be largely generalizable. The inclusion of baseline measures of attitudes toward mask wearing and the prevalence of mask use in a given country also seems very important. I believe that the presented text, as providing important information on the public acceptance of one of the key prevention methods against COVID-19, deserves to be published in Plos One.

6. PLOS authors have the option to publish the peer review history of their article (what does this mean?). If published, this will include your full peer review and any attached files.

Reviewer #1: No

Reviewer #2: No

---

## [Author Response · Author response to Decision Letter 0]

17 Sep 2021

Dear Editor Capraro,

Thank you for inviting us to submit a revised version of our manuscript “Experimental evidence that changing beliefs about mask efficacy and social norms increase mask wearing for COVID-19 risk reduction: Results from the United States and Italy.” Below we explain how we address each point raised by the reviews. In this letter we address specific comments and requests raised in your transmittal letter.

If there are any additional changes or revisions that we need to undertake, please do not hesitate to be back in contact. We look forward to hearing from you at your convenience.

Best,

Gregory A. Huber, PhD

Forst Family Professor and Chair, Political Science

Yale University

 

Response to comments and requests contained in transmittal letter:

1. Verify conformance with PLOS ONE style requirements: Done

2. Add additional information about consent to the methods section: Done. The new text is on page 9 of the revised manuscript and reads: 

The surveys received ethics approval from the Social Science, Behavioral, and Educational Research Institutional Review Board at Yale University and the Research Ethics Committee at Bocconi University. Participants were compensated for their participation, the studies did not involve deception, and written consent was obtained from each respondent prior to study enrollment.

3. Amend Funding Statement to include role of funder information and include in cover letter: Done. The amended Funding Statement should read:

“SEB and GAH acknowledge support from the Institution for Social and Policy Studies, the Center for the Study of American Politics, and the Tobin Center for Economic Policy at Yale University. PP acknowledges support from the Italian Ministry of Education Progetti di Rilevante Interesse Nazionale Grant 2017ELHNNJ. AM acknowledges support from Italian Ministry of Education Progetti di Rilevante Interesse Nazionale Grant 20177BRJXS and the European Research Council Consolidator Grant 101003183. AAM, SBO, EP, and KW were supported by Yale Institute for Global Health. The funders had no role in study design, data collection and analysis, decision to publish, or preparation of the manuscript.”

4. Remove funding-related text from manuscript: Done. We would like to use the updated Funding Statement that appears in response to #3.

5. Confirm data will be made publicly available prior to publication if manuscript is accepted: We confirm we will provide repository information prior to publication if the manuscript is accepted.

6. Include full ethics statement in methods section: Done. The revised text is described in response to #2.

7. Verify references and highlight any changes to references in cover letter: Done. In addition to reviewing the existing list of references to make sure all are up to date, we have added one reference. The new references is to the paper identified by Reviewer #1:

Barceló J, Sheen GC-H. Voluntary adoption of social welfare-enhancing behavior: Mask-wearing in Spain during the COVID-19 outbreak. PloS One. 2020;15(12):e0242764

 

Response to Reviewer #1

We thank Reviewer #1 for their support and detailed comments. We explain here how we address the two points raised in your review.

1. Engage with Barcelo and Sheen paper published in PLOS One: Reviewer #1 writes: “The authors focus the literature review on discussing prior experimental inducements of mask-wearing behavior. However, they do not incorporate some observational evidence that directly speaks to their paper. In a closely related paper published recently in PLOS ONE, Barcelo and Sheen (BS) provide observational evidence indicating “a positive correlation between a social norm of mask-wearing and mask uptake and demonstrate that uptake of facemasks is especially high among the elderly living in localities where mask-wearing behavior is popular” (1). In my opinion, the findings and interpretation of the results from this paper strongly resonate with those of BS.

Using evidence from a quota-based sample of 4,000 respondents living in Spain, BS find that mask-wearing behavior is correlated with the actual number of individuals wearing a mask in their location of residence. Quoting directly from the paper: “Table 4 reports ordinal logistic regressions that link the social acceptability of the mask-wearing behavior in a respondent’s social context and individual mask-wearing behavior after adjusting or not for demographic characteristics. The models reveal that respondents are more likely to wear a mask if they live in a region or a province where mask-wearing behavior is common. By contrast, the actual number of infected cases in a province or a region is not systematically associated with using a mask. The significant effect of social acceptability upholds after controlling for the number of infected cases in the region or province and adding fixed effect by survey day” (10-11).

Due to the resemblance of the main findings and interpretation of the results between these two papers, the authors must engage and discuss their evidence with regards to this paper in either the Introduction or the Discussion section.”

Response: We thank the reviewer for identifying this paper to us, which we now cite and discuss explicitly on page 8. We now write:

“Focusing on social distancing to reduce the spread of COVID-19, Wu and Huber [33] analyze survey data and find that individuals in the US are more likely to practice social distancing when they report individuals in their social network are also doing so after accounting for other demographics. Focusing on mask wearing, Barcelo and Sheen [36] analyze a large-scale survey conducted in Spain and show that individuals are more likely to wear a mask when rates of mask wearing are higher among those they live near, even after accounting for other demographic characteristics and COVID-19 risk factors. These observational analyses provide suggestive evidence that individuals’ own behavior is affected by the behavior of those around them, although such observational analysis is subject to standard concerns about omitted variables bias such as whether there are unmeasured interventions or covariates that explain both an individual’s behavior and the behavior of those in their social or geographic community.”

2. Clarify that treatments are descriptive, rather than injunctive, social norms: Reviewer #1 writes: “The authors may want to explicitly clarify that their paper looks at the effect of descriptive, rather than injunctive, social norms. In the authors’ social norms manipulation, the authors randomly assign respondents one of the two following conditions: “Social Norms Manipulation: Not many of the other people at the park are properly wearing a face mask OR Almost all of the other people at the park are properly wearing a face mask.” (Appendix S1).

Their experimental treatment deals with people’s perception of the prevalent behavior in their surroundings, not so much about what others approve or disapprove of. While the term “descriptive social norm” is used once in the paper, “social norm” as such is used about 20 times. I think that it is worth clarifying this concept early on to avoid confusion among readers and clarify the scope conditions of the paper.”

Response: We thank the reviewer for pointing out that our treatments manipulate descriptive rather than injunctive norms but that the text did not fully communicate this distinction. We have added new text to pages 9 that clarifies the difference between descriptive and injunctive norms: 

“We note that we use the term social norms to refer to descriptive norms, which characterize what other people are doing, rather than injunctive norms, which describe what one ought to do. Descriptive norms may undergird expectations about injunctive norms, in that people may believe they are more likely to be judged negatively for failing to follow a norm when more individuals follow the norm.” 

Additionally, throughout the manuscript we have added the word “descriptive” to the term “social norms” in several places so that the text now reads “descriptive social norms.” Finally, linking descriptive to injunctive norms, on page 22 where we discuss the effects of the treatments on the judgments of others, we have added this new text:

“This link between judgement of others and descriptive norms highlights the way in which descriptive norms may undergird injunctive norms.”

Response to Reviewer #2

We thank Reviewer #2 for their support. We did not identify any changes to make on the basis of your review.

---

## [Editor Report · Decision Letter 1]

23 Sep 2021

Experimental evidence that changing beliefs about mask efficacy and social norms increase mask wearing for COVID-19 risk reduction: Results from the United States and Italy

PONE-D-21-22869R1

Dear Dr. Huber,

We’re pleased to inform you that your manuscript has been judged scientifically suitable for publication and will be formally accepted for publication once it meets all outstanding technical requirements.

Kind regards,

Valerio Capraro

Academic Editor

PLOS ONE
---

## [Editor Report · Acceptance letter]

1 Oct 2021

PONE-D-21-22869R1 

Experimental evidence that changing beliefs about mask efficacy and social norms increase mask wearing for COVID-19 risk reduction: Results from the United States and Italy 

Dear Dr. Huber:

I'm pleased to inform you that your manuscript has been deemed suitable for publication in PLOS ONE. Congratulations! Your manuscript is now with our production department. 

Kind regards, 

on behalf of

Dr. Valerio Capraro 

Academic Editor

PLOS ONE